# Preventing Venous Thromboembolism in Ambulatory Patients with Cancer: A Narrative Review

**DOI:** 10.3390/cancers12030612

**Published:** 2020-03-06

**Authors:** Anne Rossel, Helia Robert-Ebadi, Christophe Marti

**Affiliations:** 1Division of General Internal Medicine, University Hospitals of Geneva, 1205 Geneva, Switzerland; christophe.marti@hcuge.ch; 2Faculty of Medicine, University of Geneva, 1205 Geneva, Switzerland; helia.robert-ebadi@hcuge.ch; 3Division of Angiology and Haemostasis, University Hospitals of Geneva, 1205 Geneva, Switzerland

**Keywords:** cancer associated thrombosis, VTE, venous thromboembolism, malignancy, low molecular weight heparin, direct oral anticoagulant

## Abstract

Venous thromboembolism (VTE) is frequent among patients with cancer. Ambulatory cancer patients starting chemotherapy have a 5% to 10% risk of cancer associated thrombosis (CAT) within the first year after cancer diagnosis. This risk may vary according to patient characteristics, cancer location, cancer stage, or the type of chemotherapeutic regimen. Landmark studies evaluating thrombophrophylaxis with low molecular weight heparin (LMWH) for ambulatory cancer patients have shown a relative reduction in the rate of symptomatic VTE of about one half. However, the absolute risk reduction is modest among unselected patients given a rather low risk of events resulting in a number needed to treat (NNT) of 40 to 50. Moreover, this modest benefit is mitigated by a trend towards an increased risk of bleeding, and the economic and patient burden due to daily injections of LMWH. For these reasons, routine thromboprophylaxis is not recommended by expert societies. Advances in VTE risk stratification among cancer patients, and growing evidence regarding efficacy and safety of direct oral anticoagulants (DOACs) for the treatment and prevention of CAT have led to reconsider the paradigms of this risk–benefit assessment. This narrative review aims to summarize the recent evidence provided by randomized trials comparing DOACs to placebo in ambulatory cancer patients and its impact on expert recommendations and clinical practice.

## 1. Introduction

Cancer is a major risk factor for venous thromboembolism (VTE). Approximately 20% of all VTE events are attributable to cancer [1], and an active cancer increases the risk to develop VTE up to seven times [2]. The burden of cancer associated thrombosis (CAT) is high, as the risk of mortality can be increased fourfold, depending on the type and stage of cancer [3]. Morbidity is also considerable, with an increased rate of hospitalization, home care, and decreased quality of life.

Nevertheless, the incidence of VTE in patients with cancer varies widely, ranging from 1.4% yearly [4] to over 10% [5] in high-risk patients. Among patients with pancreatic cancer receiving chemotherapy, this rate can rise as high as 20% at one year [6,7]. The relative risk for high-risk tumors is provided in Table 1. Moreover, a trend towards an increasing incidence of VTE has been observed [8], due to longer survival of patients with cancer, administration of prothrombotic chemotherapies, and improvement in the diagnosis of CAT [8]. 

Many factors influence the risk of developing CAT. In the presence of major transient risk factors, such as hospitalization for an acute medical illness, surgery, or reduced mobility, thromboprophylaxis is usually recommended, and these particular situations will not be discussed in this review [9]. However, the vast majority of CAT occur in the ambulatory setting, notably during the first 6 months following cancer diagnosis [2]. The primary location and type of cancer are important determinants of CAT risk. Breast and prostate cancer, for example, are associated with lower rates of VTE compared to pancreatic or hematologic malignancies [5]. The stage of cancer also impacts the risk of VTE [3], as well as the type of chemotherapy [10] and other supportive treatments such as red cell growth factors [11]. Metastatic disease is associated with an increased VTE risk and some treatments such as anti-EGFR molecules increase the rate of venous thromboembolic events [12,13]. As all these contributors influence the risk of VTE, the so-called “cancer population” represents in fact a highly heterogeneous population with a wide range of individual risks. 

The benefits of thromboprophylaxis using low molecular weight heparins (LMWH) in cancer patients have been largely studied in the past. Despite a reduction of the rate of VTE in various randomized controlled trials (RCTs) and meta-analyses [14,15,16], the number needed to treat (NNT) was shown to be high (40 to 50), due to the overall low rate of events. As bleeding risk is also increased in patients with cancer [17], the benefit–risk ratio of primary prophylaxis of VTE with LMWH remained uncertain. Moreover, LMWH is expensive, and subcutaneous injections are burdensome and probably more difficult for the patient to tolerate in the setting of prevention than treatment, further altering the quality of life in a population where this is a particularly important issue.

Selecting high-VTE-risk subgroups of cancer patients with a potentially more favorable benefit–risk ratio expected from thromboprophylaxis has been one of the priorities of research in this setting during the last decade [18]. In the meantime, direct oral anticoagulants (DOAC) have emerged as a potential alternative for the treatment and prevention of CAT [19,20,21]. Two recently published randomized placebo-controlled trials, AVERT [22] and CASSINI [23], evaluated rivaroxaban and apixaban for the prevention of CAT in outpatients with cancer selected as being at increased risk of VTE.

In this narrative review, we will summarize the available evidence on VTE prevention in ambulatory cancer patients, and the impact of the recent trials’ results on the latest recommendations and clinical practice.

## 2. Identification of Patients at Higher Risk of VTE

As previously discussed, thromboprophylaxis in ambulatory cancer patients is associated with a 50% reduction of VTE rate. However, the low absolute rate of events among unselected cancer patients, and treatment associated costs and potential harms result in an uncertain benefit–risk balance. Routine thromboprophylaxis is therefore not recommended in all cancer patients. Identification of cancer patients at high risk of VTE, who would thus have the highest potential benefit from thromboprophylaxis, has been the subject of active research over the last decade.

The best known risk stratification tool was derived by Khorana et al. in 2008 [18] in a cohort of 4000 ambulatory cancer patients. Two-thirds of the patients were assigned to the derivation cohort, whereas the remaining represented the validation cohort. The overall rate of VTE was low (2.2%), and the median duration of follow-up was relatively short (73 days). The multivariable regression analysis identified five predictors of VTE, which were included in the model: site of tumor (stratified by very high risk and high risk), body mass index (BMI), pre-chemotherapy hemoglobin or use of red cell growth factors, leucocyte and platelet count (Table 2). Patients with a score ≥3 were considered at high risk, corresponding to a VTE incidence of 7.1%. Of note, gastric and pancreatic cancers represented only 2% of the overall cohort.

This risk assessment model has been further validated in over 50 cohorts of ambulatory cancer patients. A recent meta-analysis identified 53 studies including more than 34,000 patients evaluating the Khorana score [24]. The reported incidence of VTE in the first 6 months was 5.0% among patients with a low-risk Khorana score (0 points), 6.6% in those with an intermediate-risk (1 or 2 points), and 11.0% in those with a high-risk Khorana score (≥3 points). The authors concluded that the Khorana score was a reliable tool to identify ambulatory cancer patients at high risk of VTE. However, only 17% of patients were classified at high risk, and most events (77%) occurred in non-high-risk patients. Using a threshold of ≥2 points to define high-risk patients, 47% of patients were classified at high risk and the incidence of VTE in this group was 8.9%. The proportion of VTE events occurring in the high-risk group was 55%, rather than 23% for a threshold of ≥3 points. Of note, the incidence of VTE in patients with a score of 0 or 1 remained substantial (5.5%). Therefore, the strength of the Khorana score lies mainly in the identification of patients with an increased risk of VTE but a low-risk score cannot put aside the occurrence of VTE.

Several adaptations of the Khorana score have been proposed and will be briefly presented below. The modifications included adding additional variables such as D-dimers and P-selectin [25]; treatment by gemcitabine or platinum [26]; replacing BMI by functional status [27], or adding metastatic disease, vascular compression, or previous VTE [28] (Table 2). In addition, novel scores have also been proposed including other predictive factors [29]. The proportion of patients classified as high risk by these different scores and the corresponding VTE incidence are summarized in Table 3.

After having shown that high levels of D-dimer and P-selectin [30,31] were associated with an increased risk of VTE, Ay et al. elaborated a VTE-risk assessment tool including these biomarkers, in the Vienna-Cancer And Thrombosis study (Vienna-CATS) cohort. The six-month VTE risk was 17.7% in the high-risk group (≥3 points). Although the positive predictive value of this score appeared higher than the Khorana score, its widespread clinical use is hampered by requirement of specific biomarkers, such as P-selectin. More recently, the same group elaborated a new score with only two items: tumor site and of D-dimer levels before chemotherapy, and reported [32] better c-indexes than the Khorana score in both the derivation (0.66 vs. 0.61) and validation (0.68 vs. 0.56) cohorts. 

Based on data from the PROTECHT study [10], Verso et al. proposed to add gemcitabine, cisplatine, or carboplatin as additional risk factors [26]. Pelzer et al. suggested an adaptation of the Khorana score by replacing BMI with performance status in the CONKO004 study [27] evaluating LMWH prophylaxis in patients with pancreatic cancer [33]. This score has, to date, not been externally validated.

The ONKOTEV score [28] includes the presence of metastatic disease, the compression of vascular structures, and history of previous VTE (Table 2). In the validation study, the 12-months probability of VTE (including incidentally diagnosed VTE) was 33.9% among patients with a score of 3 or more, 19.4% among patients with a score of 2, 9.7% among patients with a score of 1, and 3.7% among patients with a score of 0. The AUC was reported as higher for ONKOTEV than Khorana score at 6 months (0.75 vs. 0.59) in this cohort. This score has been further validated on a retrospective cohort of patients with pancreatic cancer [34].

The COMPASS-CAT score was elaborated to improve the assessment of VTE risk for patients with lung, colon, breast, and ovarian cancers [29]. This score includes patient-related (cardiovascular risk factors, history of VTE, platelet count, recent hospitalization), cancer-related (stage, time since diagnosis), and treatment-related items (anthracycline or anti-hormonal therapy, central venous catheter). The risk of VTE was 1.7% in the low/intermediate-risk group, and 13.3% in the high-risk group. The score had a good discriminatory capacity (AUC 0.85), but external prospective validation is lacking. 

A comparative analysis of the performance of these predictive models (with the exception of the Vienna-CATS and COMPASS-CAT score due to the lack of biomarker measurement and cardiovascular risk assessment) has been performed in a cohort of 776 ambulatory patients receiving chemotherapy [24]. Overall, the discriminatory power of the scores was low, with a c-index of approximately 0.60 for all scores. A positivity threshold of 2 points improved performance of all scores and captured a higher proportion of VTE. Another comparison of several scores [35] was conducted on a prospective multinational cohort of 876 patients with various cancers. C-statistics of the scores were again low, ranging from 0.52 for Khorana to 0.59 for PROTECHT. 

In conclusion, although some alternative risk assessment models have a superior reported positive predictive value or overall accuracy in the studied cohorts, the Khorana score remains the most widely validated prediction score to date. Using a threshold of 2 or more, the Khorana score allows identifying a subgroup of cancer patients at high risk of VTE (expected 6-months VTE incidence of around 9%), representing potential candidates for thromboprophylaxis with a potentially favorable risk–benefit ratio. 

## 3. Evidence Regarding Primary Prophylaxis

Many studies have been conducted on the use of anticoagulants for primary prevention of VTE in patients with cancer. Most studies used LMWH in this setting as this class of anticoagulants has been shown to be superior to Vitamin K antagonists (VKA) for the treatment of cancer associated thrombosis [21]. LMWH enhance antithrombin action to inhibit factor Xa. Their pharmacokinetic is better predictable than unfractionated heparin but their renal metabolism precludes their use in renal insufficiency. A few studies evaluated the use of vitamin K antagonists as prophylactic treatment. These molecules inhibit the synthesis of vitamin K-dependent coagulation factors, and their efficacy may be subject to important variations depending on vitamin K intake, especially in cancer patients receiving chemotherapy often resulting in reduced oral intake and/or nausea and vomiting. A selection of randomized studies using LMWH is reported in Table 4. 

The primary outcome of earlier studies was the effect of LMWH on survival, after some encouraging in vitro and in vivo results [36]. However, the hoped benefit of LMWH on survival in cancer patients could not be demonstrated in large scale studies [37,38]. Thereafter, VTE incidence was the main outcome. 

The SAVE-ONCO study [15] randomized 3212 patients with metastatic or locally advanced solid cancers to receive semuloparin or placebo, regardless of their thrombotic risk. Almost 70% had metastatic disease. Patients who received semuloparin presented fewer thrombotic events (HR 0.36; 95%CI 0.21–0.60), without a significant difference in the rate of major bleeding (HR 1.05; 95%CI 0.55–1.99). The benefit was particularly important among patients with lung and pancreatic cancers, with a relative VTE risk reduction of 64% (RR 0.36, 95% CI 0.17–0.76) and 78% (RR 0.22, 95% CI 0.06–0.74), respectively. However, the absolute risk reduction in the overall population of patients was low (2.2%). 

The PROTECHT study [14] compared nadroparin to placebo in 1150 patients with metastatic or locally advanced cancer of various origins, without cerebral metastasis. Treatment was initiated for the duration of chemotherapy or 4 months. The primary efficacy outcome was a composite including VTE, arterial events (acute myocardial infarction, ischemic stroke, acute arterial thromboembolism), and VTE-related death. Whereas nadroparin significantly decreased the incidence of the composite outcome, the effect on VTE incidence was non-significant. (RR 0.50; 95%CI 0.22–1.13) and there was a trend towards more bleeding events (RR 5.46; 95%CI 0.30–98.43). Despite an inclusive definition of thromboembolism, the overall number of events was low, even in the placebo group where the occurrence of VTE was lower than the rate reported observational studies among patients treated by chemotherapy [7] (2.9% vs. 7.3% at 3.5 months). A possible explanation could be that the treatment duration and follow-up were relatively short, (median 112 days). Moreover, mortality at the end of treatment was low (4.3% vs. 4.2%), reflecting the selection of patients with a better prognosis than the general oncologic population. 

Haas et al. [39] compared certoparin to placebo over 6 months in patients with metastatic breast cancer or stage III/IV non-small cell lung carcinoma. No significant difference was found in the rate of VTE (RR 0.57; 95%CI 0.24–1.35) or major bleeding (1.12; 95% CI 0.52–2.38). 

The FRAGMATIC trial was conducted among patients with primary bronchial carcinoma of any stage [40], comparing dalteparin to placebo. VTE was less frequent in the LMWH group (RR 0.57; 95%CI 0.42–0.77), without an increase in major bleeding (RR 1.50; 95%CI 0.62–3.66). However, only 18.4% of patients were fully compliant, and 39% received half of the planned syringes or less.

In patients receiving gemcitabine for pancreatic cancer, adding primary prophylaxis with therapeutic doses of dalteparin significantly reduced VTE or arterial events (RR 0.15, 0.04–0.61) [41]. Despite therapeutic doses, the rate of bleeding events was low without a significant difference between groups (3.4% vs. 3.2%). In this study, VTE was a significant predictor of mortality (HR 1.93, 95% CI 1.23–3.03) but LMWH had no effect on mortality. Another randomized controlled study comparing enoxaparin added as primary prophylaxis to chemotherapy versus chemotherapy alone in patients with advanced pancreatic cancer (CONK004) [33] also showed a 3 month decrease in VTE risk with enoxaparin (HR 0.12; 95% CI 0.03–0.52). The rate of VTE in the control group (15% at 3 months) was remarkably high in this study. There was no significant increase in major bleeding (HR 1.4, 0.35–3.72). 

LMWH primary prophylaxis trials in the setting of cancer are thus highly heterogeneous in terms of study populations, as some included unselected populations of cancer patients and others a very specific subgroup of patients with high-risk advanced cancer. This heterogeneity is well reflected by the event rates in the placebo (or no anticoagulation) arms (Table 4). In the two large placebo-controlled randomized SAVE-ONCO and PROTECHT studies of unselected cancer patients, the VTE rate in the placebo arm was 3.4% and 2.9%, respectively, indicating a low VTE risk cancer population. 

Khorana et al. [42] aimed to assess LMWH prophylaxis in a selected population of patients with high risk of thrombotic event, defined by a Khorana score of ≥ 3 points. This study terminated prematurely (98 patients) because of a poor accrual. A non-significant reduction of the rate of VTE was observed (12% in dalteparin group vs. 21% in control arm; HR 0.69, 95% CI 0.23–1.89). In a phase II study, Zwicker et al. [43] stratified patients depending of their level of circulating tissue factor-bearing micro particles (TFMP) and randomized those at higher risk to enoxaparin or standard treatment. In this small study (*n* = 34), the rate of VTE at 2 months was particularly high in the control group (27.3%) and was significantly decreased in the enoxaparin arm (HR 0.15; 95% CI 0.03–0.97).

Several studies also evaluated Vitamin K Antagonists (VKA) prophylaxis in patients with cancer. The impact on VTE and survival were inconstant while some increase in bleeding risk was reported [44,45,46,47]. Finally, several studies evaluated thromboprophylaxis in specific subgroups of cancers such as multiple myeloma patients receiving thalidomide and derivatives. Among these patients, the one-year VTE risk may increase over 20% and may be reduced by the administration of aspirin LMWH or VKA [48,49]. 

A Cochrane meta-analysis [16] published in 2016 included all randomized controlled trials comparing any anticoagulant to placebo or other anticoagulant, in outpatients receiving chemotherapy. Overall, primary thromboprophylaxis with LMWH significantly reduced the incidence of symptomatic VTE in outpatients treated with chemotherapy (RR 0.54; 95%CI 0.38–0.75). There was a trend towards increased major bleeding with LMWH, but this result did not achieve statistical significance (RR 1.44; 95%CI 0.98–2.11). Despite the clear benefit in terms of VTE risk reduction, and the relative safety regarding adverse bleeding events, the systematic use of LMWH as a prophylactic treatment has not been recommended, mainly because the absolute risk reduction remains limited in unselected populations of cancer patients. Moreover, the burden of daily subcutaneous injections is substantial, and premature treatment interruptions occurred in a large proportion of participants even in the setting of RCTs [40,49]. 

## 4. Use of DOAC for VTE Prevention

DOAC act by direct inhibition of factor Xa (rivaroxaban, apixaban, edoxaban) or factor IIa (dabigatran). Their major advantage is oral administration without requiring monitoring. However, because of their cytochrome-dependent metabolism, they are subject to potential drug–drug interactions. Andexanet alfa, a recombinant variant of human factor Xa, competes with endogenous factor Xa and has been shown efficient to decrease anti-factor Xa activity and restore hemostasis. Andexanet alfa is usually administered using a 400 mg bolus administered in 15 min followed by a 480 mg infusion over 2 h for patients receiving apixaban (800 mg bolus over 30 min followed by a 960 mg infusion for those receiving edoxaban or rivaroxaban) [52].

Two major phase III trials have recently been published, assessing apixaban [22] and rivaroxaban [23] for preventing VTE in ambulatory patients with cancer at high risk of VTE (Khorana score ≥ 2).

In the AVERT study [22], 574 ambulatory cancer patients from 13 centers starting a new course of chemotherapy with a Khorana score ≥ 2 were randomized to apixaban 2.5mg twice daily or placebo for 180 days. Around 25% of patients had lymphoma, 25% a gynecologic cancer, whereas pancreatic cancer was present in 13% (Table 5). Among solid cancers, one quarter were metastatic. Two thirds of the participants had a Khorana score of 2, and the remaining were ≥3. The primary efficacy outcome was VTE, including proximal DVT of upper or lower extremities, PE (symptomatic or incidental), or VTE-related death at 210 days. VTE occurred in 12/488 (4.2%) patients allocated to apixaban and 28/275 (10.2%) patients allocated to placebo (HR 0.41; 95% CI 0.26–0.65). Major bleeding occurred in 10/288 (3.5%) patients allocated to apixaban and 5/275 (1.8%) in the control group (HR 2.00; 95%CI 1.01–3.95). The increase in major bleeding was mainly due to higher rates of mucous bleedings, especially in the gastro-intestinal (GI), urinary, and gynecological tracts, and most events occurred in patients who entered the study with cancers in these locations. There was no significant difference in clinically relevant non-major bleeding (CRNMB).

In the CASSINI study [23], 841 ambulatory cancer patients from 11 countries with a solid tumor or lymphoma starting a new chemotherapy with a Khorana score ≥ 2 were randomized to rivaroxaban 10mg daily versus placebo over 180 days. The study population included patients with advanced metastatic cancers of different origins. Patients with primary brain cancer or cerebral metastases, and patients with hematological malignancies were excluded. One third of patients had pancreatic cancer, 21% an upper GI tract cancer, 15% lung cancer, 8% gynecological cancers, and 7% lymphomas. Among those with solid cancers, 54.5% had a metastatic disease.

Systematic screening for lower limb deep vein thrombosis (DVT) was performed with compression ultrasound (CUS) and only patients without DVT were included. Of note, 4.5% of patients screened with CUS had DVT and were excluded. Systematic lower limb CUS was repeated at 8, 16, and 24 weeks. The primary outcome was the composite of symptomatic or screen-detected proximal lower extremity DVT, symptomatic or incidental PE, symptomatic DVT in upper limb, distal DVT in lower limb, or VTE-related death. The primary outcome occurred in 25 of 420 patients (6.0%) in the rivaroxaban group and 37 of 421 (8.8%) in the control group (HR 0.66, 95%CI 0.40–1.09). A secondary analysis restricted to the on-treatment period showed a VTE rate of 2.6% on rivaroxaban versus 6.4% on placebo (HR 0.40, 95%CI 0.20–0.80). Major bleeding occurred in eight of 405 patients (2%) in the rivaroxaban group and in four of 404 (1%) in the control group (HR 1.96; 95%CI 0.59–6.49). There was no significant difference in clinically relevant non-major bleeding (CRNMB).

In summary, in AVERT, apixaban reduces VTE at the expense of increased major bleeding. In CASSINI, rivaroxaban does not significantly reduce VTE but does not significantly increase major bleeding. The differences in interventions, outcome definitions, and populations (Table 5) impact the direct comparison of results from AVERT and CASSINI trials. 

A pooled analysis of the two studies has nevertheless been performed and showed a 6-month VTE risk reduction of 0.56 (95%CI 0.35–0.89) on DOACs, with a non-significant increase in major bleeding (1.96; 95%CI 0.80–4.82) [53]. In terms of absolute difference, this corresponded to a VTE risk reduction of 4% (95%CI 0.01–0.07, NNT 25) at the cost of a 1% (95%CI 0.0–0.02) increase (albeit statistically non-significant) in major bleeding (NNH 100). This risk–benefit ratio compares favorably with previous studies using LMWH, and all the more so when taking into account the lower cost and easier route of administration of DOACs compared to LMWH. However, this more favorable balance seems mainly to be due to the selection of patients with an higher basal VTE risk as the observed VTE relative risk reduction is very similar in studies using DOACs (0.56; 95%CI 0.35–0.89) [53] or LMWH (0.54; 95%CI 0.38–0.75) [16].

Based on these studies, thromboprophylaxis using apixaban or rivaroxaban has been endorsed in recent recommendations by expert societies. The American Society for Clinical Oncology (ASCO) recommends thromboprophylaxis in ambulatory cancer patients with a Khorana score ≥ 2 (moderate strength of recommendation) [54]. The International Initiative on Thrombosis and Cancer (ITAC) and International Society for Thrombosis and Hemostasis (ISTH) recently recommended thromboprophylaxis using apixaban or rivaroxaban in ambulatory patients receiving chemotherapy at intermediate-to-high risk of VTE based on cancer type or a validated risk assessment model [55]. According to these guidelines, patients with locally advanced or metastatic pancreatic cancer are considered at high risk of VTE, regardless of their score and thromboprophylaxis is recommended in these patients in the absence of a high risk of bleeding. ASCO and ISTH-ITAC guidelines are provided in Table 6.

Several issues remain to be highlighted. First, selection of patients using the Khorana score ≥ 2 resulted in a higher rate of VTE (10% in the placebo arm in AVERT and 9% in CASSINI) compared to unselected series of patients, which confirms that this score can be used as a prediction tool in this setting. Second, the tendency of DOAC to be associated with higher risk of mucosal bleeding seems once again to be confirmed and these molecules should be used with caution in patients with GI cancer. Third, VTE events in AVERT and CASSINI trials included incidentally diagnosed VTE (incidental PE represented one fourth of all events in both studies), whereas the necessity to treat these events is not fully elucidated. Fourth, systematic screening for DVT was performed before inclusion in the CASSINI study; this “pre-selection” of patients without DVT may have influenced the results. Moreover, systematic CUS was also performed in CASSINI, and asymptomatic proximal DVTs contributed to 29% of all events in the placebo arm. As the evolution of these DVTs, had they been undiagnosed, is unknown, the influence on the primary outcome is uncertain. Finally, an important point to highlight is the very high rate of discontinuation of treatment (37% in AVERT and 47% in CASSINI) reflecting the complexity of care in patients with active cancer on chemotherapy.

## 5. Conclusions

Thromboprophylaxis in ambulatory cancer patients using LMWH or DOACs (apixaban or rivaroxaban) reduces VTE events by about one half, but with a potential increase in major bleeding. DOAC represent an interesting option because of their oral administration and lower costs compared to LMWH. Large scale thromboprophylaxis prescription in ambulatory cancer patients is however not advised, and selection of the patients at high VTE risk without being at high risk of bleeding remains the main challenge. Patients with cancers at very high VTE risk (e.g., pancreas) are most likely to benefit most from primary prophylaxis with DOACs, whereas caution is needed in patients with GI and genitourinary cancers. Further studies based on specific cancers populations or alternative risk assessment models may allow to further improve patient selection. In the complex setting of patients with active cancer on chemotherapy, the decision to initiate thromboprophylaxis should be discussed individually, taking into account tumor site, concomitant treatments, bleeding risk, and most importantly patient’s values and preferences.

## Figures and Tables

**Table 1 cancers-12-00612-t001:** Relative risk of thromboembolism according to cancer type, compared to general population (based on [5]).

Cancer Site	Incidence Rate Ratio (IRR) (95%CI)
Overall	3.96 (3.66–4.27)
Pancreas	15.56 (10.50–23.06)
Hematological	12.65 (10.04–15.94)
Brain	10.40 (5.48–18.08)
Lung	7.27 (5.93–8.91)

**Table 2 cancers-12-00612-t002:** Risk assessment scores.

Patients Characteristics	Khorana Score [18]	CATS Score [25]	PROTECHT Score [26]	CONKO Score [27]	ONKOTEV Score [28]
Pancreatic or gastric cancer	+2	+2	+2	+2	-
Lung, gynecologic, or genitourinary cancer (except prostate), or lymphoma	+1	+1	+1	+1	-
Hemoglobin < 10 g/dL* or use of red cell growth factors	+1	+1	+1	+1	-
White blood cell count > 11 × 10^9^/L*	+1	+1	+1	+1	-
Platelet count ≥ 350 × 10^9^/L*	+1	+1	+1	+1	-
Body mass index > 35 kg/m^2^	+1	+1	+1	-	-
D-dimers ≥ 1.44 μg/mL*	-	+1	-	-	-
P-selectin ≥ 53.1 ng/mL*	-	+1	-	-	-
Gemcitabine or platinum chemotherapy	-	-	+1	-	-
WHO performance status ≥ 2	-	-	-	+1	-
Khorana score ≥ 2 points	-	-	-	-	+1
Metastatic disease	-	-	-	-	+1
Previous venous thromboembolism	-	¬-	-	-	+1
Vascular/lymphatic macroscopic compression	-	-	-	-	+1
	High risk ≥3 Intermediate risk 1–2 Low risk 0				

* values measured before the beginning of chemotherapy.

**Table 3 cancers-12-00612-t003:** Incidence of venous thromboembolism (VTE) in patients classified as high risk according to different prediction models.

Score and Threshold for Defining High Risk	Incidence of VTE in the High-Risk Category	Proportion of Patients Classified in the High-Risk Category	Follow-Up
Khorana ≥ 3	11% [24]	17%	6 months
CATS ≥ 3	17.7% [25]	25.7%	6 months
PROTECHT ≥ 3	8.1% [26]	32%	12 months
COMPASS ≥ 7	13.3% [29]	50.5%	12 months
ONKOTEV ≥ 2	33.9% [28]	7%	12 months
Khorana ≥ 2	8.9% [24]	47%	6 months

**Table 4 cancers-12-00612-t004:** Randomized studies on VTE prevention with low molecular weight heparin (LMWH).

Author (Year)	Type of Cancer	Stage of Cancer (Proportion Metastatic)	Drug	Patient Number	Treatment Duration	Outcome Definition	VTE Relative Risk (95%CI)	Major Bleeding RR (95%CI)	Event Rate in Control Group
Agnelli (2012) [15]	Lung, pancreas, stomach, colon, rectum, bladder, ovary	Metastatic (68%) or locally advanced	Semuloplasmin 20 mg/d	3214	3 m	VTE or VTE death	0.36 (0.21–0.60)	1.05 (0.55–1.99)	3.4%
Agnelli (2009) [14]	Lung, GI, pancreatic, breast, ovarian, head, neckNo brain metastasis	Metastatic (unknown) or locally advanced	Nadroparin 3800 UI sc/d	1150	120 d	Composite including VTE, arterial TE or VTE death	0.5 (0.22–1.13)	5.46 (0.30–98.4)	2.9%
Haas (2012) [39]	Breast or non-small cell lung cancerNo brain metastasis	Metastatic breast cancer, stage III–IV lung cancer	Certoparin 3000 IU sc/d	883	6m	Objectively confirmed symptomatic or asymptomatic VTE	0.57 (0.24–1.35)	2.19 (0.89–5.70)	3.1%
Kakkar (2004) [37]	Breast, lung, GI, pancreas, liver, genitourinary	Metastatic (84%) or locally advanced	Dalteparin 5000 UI sc/d	374	1 y	Symptomatic confirmed VTE*	0.77 (0.21–2.84)	2.91 (0.12–70.9)	2.7%
Klerk (2005) [50]	Solid tumor	Metatastic (91%) or locally advanced	Nadroparin bid over 14d, then od	302	6w	NA	NA	5.20 (0.62–44.0)	NA
Macbeth (2016) [40]	Bronchial carcinoma	All stages, metastatic (61%)	Dalteparin 5000IU sc/d	2202	6m	NA	0.57 (0.42–0.77)	1.50 (0.62–3.66)	9.7%
Maraveyas (2012) [41]	Pancreatic cancer	Metastatic (54%) or locally advanced	Dalteparin 200 UI/kg sc od for 4w, then 150 UI/kg	123	12w	VTE or arterial event	0.15 (0.04–0.61)	1.05 (0.15–7.22)	18.3%
Pelzer (2015) [33]	Pancreatic cancer	Metastatic (76%) or locally advanced	Enoxaparin 1 mg/kg od	312	3 m	VTE or arterial event	0.12 (0.03–0.52)	1.4 (0.35–3.72)	14.5%
Perry (2010) [51]	Stage 3 or 4 glioma	Locally advanced	Dalteparin 5000 IU sc/d	186	6 m	VTE or arterial event	0.51 (0.19–1.4)	4.2 (0.48–36)	14.9%
Van Doormaal (2011) [38]	Stage IIIb non-small cell pulmonary carcinoma, prostate, pancreatic cancer	Metastatic (32%)	Nadroparin bid over 14d, then half therapeutic dose	503	Median duration: 12.6w	VTE	1.12 (NA)	1.18 (0.49–2.85)	6.5%

GI: gastro-intestinal, sc: subcutaneous, od: once daily, bid: Bi-daily, d: days, w: weeks, m: months, VTE: venous thromboembolism, NA: not available.

**Table 5 cancers-12-00612-t005:** Characteristics of the AVERT and CASSINI trials.

Study Characteristics	AVERT	CASSINI
Intervention	Apixaban 2 × 2.5 mg/d	Rivaroxaban 10 mg/d
Type of cancer	Lymphoma 25%, gynecologic 26%, pancreas 13%, lung 10%	Pancreas 33%, upper GI 21%, lung 15%, lymphoma 7%
Outcome definition	Symptomatic or incidental VTE	Symptomatic or incidental VTE or VTE death *
VTE rate in control group	10.2%	8.8%
Mortality in control group	9.8%	23.8%

* Systematic DVT screening, VTE: venous thromboembolism.

**Table 6 cancers-12-00612-t006:** Recommendations for thromboprophylaxis in ambulatory patients with cancer.

ASCO [54]	ISTH-ITAC [55]
Routine thromboprophylaxis should not be offered to all outpatients with cancer	Primary prophylaxis in ambulatory patients receiving systemic cancer therapy is not recommended routinely
High-risk patients with cancer and Khorana score ≥ 2 may be offered thromboprophylaxis with apixaban, rivaroxaban, or LMWH in the absence of risk factors for bleeding	Primary prophylaxis with LMWH is indicated in ambulatory patients with locally advanced or metastatic pancreatic cancer treated with systemic cancer therapy and who have a low risk of bleeding
Patients with multiple myeloma receiving thalidomide or lenalidomide should receive thromboprophylaxis with AAS or LMWH for lower-risk patients and LMWH for higher-risk patients	Primary prophylaxis with DOAC (rivaroxaban or apixaban) is recommended in outpatients receiving systemic anticancer therapy at intermediate-to-high risk of VTE, identified by cancer type (i.e., pancreatic) or by a validated risk assessment model (i.e., a Khorana score ≥2), and not at a high risk of bleeding

AAS: aspirin, LMWH: low-molecular weight heparin, DOACS: direct anticoagulants.

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
