# Peer review of "Preventing Venous Thromboembolism in Ambulatory Patients with Cancer: A Narrative Review"

_cancers, 2020, doi:10.3390/cancers12030612_

Round 1

Reviewer 1 Report

A nice and concise review on an important topic that will impact patient care. A few comments and suggestions to improve the manuscript. 

  1. A section should be dedicated to the description of various anticoagulants available for prophylaxis for cancer patients. This section does not need to be extensive, but a couple of lines regarding the mechanism of action, potential adverse events, contraindications and reversal strategies. Although there is no study to elucidate the pros and cons of using reversal with andexanet alfa, the authors can mention a possible reversal strategy that is followed at their institution. 
  2. A special comment should be made regarding the adverse effects of using coumadin in cancer patients- the small but important risk of inducing calciphylaxis. 
  3. Please mention the risk prevention strategy for multiple myeloma patients on thalidomide or lenalidomide. 
  4. A table enlisting potentially high-risk tumors for VTE can be added. 

Minor comments

  1. line 48-49- The study with VEGF only pointed out the increased risk of arterial thrombosis, whereas the EGFR study talked about venous thrombosis. A factual correction would help the reader understand this better. 
  2. line 93-97: Some clarification needed- Did 45% VTE events happened in patients with Khorana score 0-1? How is it we are calling Khorana score to be good in identifying high-risk patients.
  3. Line 163-169: PROTECHT study should be discussed in a little bit more detail with the limitations of the study. Lines 160-170, the authors write, that the patients were 'low-risk' for VTE. Is it their assumption or they are citing a source. I believe, the researchers involved in PROTECHT study do not list this as a limitation.
  4. CONK004 trial is not mentioned.
  5. Finally, a table or a flow chart should be made to help the reader understand how to approach a patient with a new diagnosis of cancer and suggest thromboprophylaxis.

Author Response

Point by point answer to reviewers’ comments

Please find below the point by point answer to reviewers ‘comments. Reviewers ‘comments appear in black plain text and authors ‘responses in italic.

Reviewer 1

  1. A nice and concise review on an important topic that will impact patient care. A few comments and suggestions to improve the manuscript. 

We thank the reviewer for this supporting comment.

  1. A section should be dedicated to the description of various anticoagulants available for prophylaxis for cancer patients. This section does not need to be extensive, but a couple of lines regarding the mechanism of action, potential adverse events, contraindications and reversal strategies.

We thank the reviewer for this suggestion. A paragraph describing the main characteristics of LMWH and VKA has been added to section 3 (page 5 lines 170-175). The mechanism of action and reversal strategies for DOAC has been added to the dedicated section. (page 8, lines 257-260)

  1. Although there is no study to elucidate the pros and cons of using reversal with andexanet alfa, the authors can mention a possible reversal strategy that is followed at their institution.

In order to provide a possible strategy for reversal of factor Xa inhibitors, the doses and mode of administration of andexanet alfa have been added to the section dedicated to DOACs. (page 8 lines 260-264). We are not certain that this addition contributes to the clarity of the manuscript but we rely on the editor’s decision for this point.

  1. A special comment should be made regarding the adverse effects of using coumadin in cancer patients- the small but important risk of inducing calciphylaxis.

We thank the reviewer for this comment about this rare but serious complication of vitamin K antagonists use. As our manuscript is mainly dedicated to LMWH or DOAC thromboprophylaxis strategies, and given the paucity of studies evaluating VKA prophylactic strategies and the rarity of these complications we would suggest not to mention this aspect in order not to make our manuscript more cumbersome.

  1. Please mention the risk prevention strategy for multiple myeloma patients on thalidomide or lenalidomide. 

We’ve added a paragraph on multiple myeloma in section 2, regarding evidence on prophylactic anticoagulation (Page 6 Lines 238-241)

  1. A table enlisting potentially high-risk tumors for VTE can be added. 

Thank you for this suggestion. A table describing the association between some type of tumors and VTE risk has been added to the manuscript as suggested. (Table 1 page 2).

Minor comments

  1. line 48-49- The study with VEGF only pointed out the increased risk of arterial thrombosis, whereas the EGFR study talked about venous thrombosis. A factual correction would help the reader understand this better. 

We thank the reviewer for this precision. In order to clarify this point, the type of thrombo-embolic event induced by anti-EGFR treatment has been added to the manuscript. (Page 2 Lines 52).

  1. line 93-97: Some clarification needed- Did 45% VTE events happened in patients with Khorana score 0-1? How is it we are calling Khorana score to be good in identifying high-risk patients.

As discussed in the manuscript, the main advantage of Khorana score is its extensive validation and its utilization in two recent impact studies evaluating DOAC thromboprohylaxis. However, the discriminative performance of this score remains modest as illustrated by its modest AUC in comparative studies and patients with Khorana scores 0 to 1 remain at increased risk of VTE compared to the general population, although this absolute risk is usually considered insufficient to justify thromboprophylaxis. The sentence about the proportion of VTE events in the non-high-risk groups according to Khorana score has been modified in order to clarify this aspect. (Page 3 Lines 101-106).

  1. Line 163-169: PROTECHT study should be discussed in a little bit more detail with the limitations of the study.

As discussed in the manuscript, the main limitation in our view of the PROTECHT study was the relatively low risk of VTE events (11/381, 2.9%) in the placebo group. As suggested by the reviewer, additional information regarding follow-up duration, and clarifications regarding the primary composite outcome (including arterial events) and VTE events have been added to the manuscript (Page 5 Lines 188-200).

  1. Lines 160-170, the authors write, that the patients were 'low-risk' for VTE. Is it their assumption or they are citing a source. I believe, the researchers involved in PROTECHT study do not list this as a limitation.

Please refer to the answer to point 3. As discussed above, the rate of VTE events (excluding arterial events) in the placebo arm of the PROTECHT trial was 11/381(2.9%) at four months, which limits the precision of treatment effect estimate on this outcome. This rate of events may be considered as relatively low compared to other cohorts of cancer patients in observational or interventional studies. We acknowledge that the use of the term “low-risk” may be misleading and we corrected the paragraph accordingly. (page 5 Lines 188-200). A reference reporting VTE events rate in observational studies has been added to this paragraph.

  1. CONK004 trial is not mentioned.

We thank the reviewer for this comment. The CONK004 trial was actually discussed (ref 33) but its acronym was not mentioned. In order to clarify this point, the CONK004 acronym has been added to the related sentence. (page 6 Line 215).

  1. Finally, a table or a flow chart should be made to help the reader understand how to approach a patient with a new diagnosis of cancer and suggest thromboprophylaxis.

We thank the reviewer for this suggestion. We agree that a table might be useful for the readers. An additional table summarizing the main recommendations for thromboprophylaxis by expert societies has been added to the manuscript (table 6 page 9).

Reviewer 2 Report

This review entitled “Preventing venous thromboembolism in ambulatory patients with cancer : a narrative review ” summarizes available evidence regarding risk assessment models to predict venous thromboembolism in cancer patients and the benefit-risk ratio of primary thromboprophylaxis in ambulatory patients with cancer receiving chemotherapy, including recent RCTs evaluating the efficacy  and safety of DOACs in this setting.

The manuscript is well written and extensively describes available data on this topic.

However, I have some minor concerns regarding the manuscript that should be addressed by the authors :

-In the abstract, the authors state that “ambulatory cancer patients starting chemotherapy have a 5 to 10% risk of cancer associated thrombosis”; however, in some studies, the cumulative incidence of VTE is higher and reaches approximatively 20% in pancreatic cancer patients

-In the introduction section, line 36-37, same comment.

In a retrospective analysis of the United States IMPACT health care claims database, patients with a range of solid tumors who started chemotherapy, the overall incidence of VTE 12 months after starting chemotherapy was 13.5% at 12 months (range 9.8%-21.3%) with the highest risk for VTE identified in patients with pancreatic cancer.

Reference: Lyman GH, Eckert L, Wang Y, Wang H, Cohen A. Venous thromboembolism risk in patients with cancer receiving chemotherapy: a real-world analysis. Oncologist. 2013;18(12):1321-9. doi: 10.1634/theoncologist.2013-0226.

Similarly, in a recent prospective, observational multicenter study of pancreatic cancer patient, the cumulative probability of VTE was 19.21% (95% CI 16.27-22.62) at 12 months

Reference: Frere C, Bournet B, Gourgou S, Fraisse J, Canivet C, Connors JM, Buscail L, Farge D; BACAP Consortium. Incidence of Venous Thromboembolism in Patients with Newly Diagnosed Pancreatic Cancer and Factors Associated With Outcomes. Gastroenterology. 2019 Dec 13. pii: S0016-5085(19)41921-5. doi: 10.1053/j.gastro.2019.12.009. [Epub ahead of print]

The authors should cite both references and highlight the high risk for VTE in pancreatic cancer patients.

-In the introduction section, line 48, please add that the use of red cell growth factors has also been associated with an increased risk of VTE

- in the introduction section, line 81, please add “pre-chemotherapy  haemoglobin or use of red cell growth factors’ in this item of the Khorana score

-in the paragraph “use of DOACs for VTE prevention”, line 275 the authors cite the most recent guidelines on the management of VTE in cancer patients (ASCO and ITAC). Please add that the ITAC guidelines also recommend the use of primary thromboprophylaxis in patients with locally advanced or metastatic pancreatic cancer in the absence of a high risk of bleeding.

Author Response

Point by point answer to reviewers’ comments

Please find below the point by point answer to reviewers ‘comments. Reviewers ‘comments appear in black plain text and authors ‘responses in italic.

Reviewer 2

  1. This review entitled “Preventing venous thromboembolism in ambulatory patients with cancer: a narrative review ” summarizes available evidence regarding risk assessment models to predict venous thromboembolism in cancer patients and the benefit-risk ratio of primary thromboprophylaxis in ambulatory patients with cancer receiving chemotherapy, including recent RCTs evaluating the efficacy and safety of DOACs in this setting.

The manuscript is well written and extensively describes available data on this topic.

We thank the reviewer for this supporting comment.

  1. However, I have some minor concerns regarding the manuscript that should be addressed by the authors :In the abstract, the authors state that “ambulatory cancer patients starting chemotherapy have a 5 to 10% risk of cancer associated thrombosis”; however, in some studies, the cumulative incidence of VTE is higher and reaches approximatively 20% in pancreatic cancer patients

-In the introduction section, line 36-37, same comment.

In a retrospective analysis of the United States IMPACT health care claims database, patients with a range of solid tumors who started chemotherapy, the overall incidence of VTE 12 months after starting chemotherapy was 13.5% at 12 months (range 9.8%-21.3%) with the highest risk for VTE identified in patients with pancreatic cancer.

Reference: Lyman GH, Eckert L, Wang Y, Wang H, Cohen A. Venous thromboembolism risk in patients with cancer receiving chemotherapy: a real-world analysis. Oncologist. 2013;18(12):1321-9. doi: 10.1634/theoncologist.2013-0226.

Similarly, in a recent prospective, observational multicenter study of pancreatic cancer patient, the cumulative probability of VTE was 19.21% (95% CI 16.27-22.62) at 12 months

Reference: Frere C, Bournet B, Gourgou S, Fraisse J, Canivet C, Connors JM, Buscail L, Farge D; BACAP Consortium. Incidence of Venous Thromboembolism in Patients with Newly Diagnosed Pancreatic Cancer and Factors Associated With Outcomes. Gastroenterology. 2019 Dec 13. pii: S0016-5085(19)41921-5. doi: 10.1053/j.gastro.2019.12.009. (Epub ahead of print)

The authors should cite both references and highlight the high risk for VTE in pancreatic cancer patients.

Thank you for this comment. the average reported range for VTE events at one year in ambulatory cancer patients is around 10%[1] but varies widely according to VTE definition (systematic screening, the inclusion of asymptomatic or arterial events etc…), cancer stage and location and the type of chemotherapeutic regimen. We acknowledge that this risk may be higher in selected cancer populations such as multiple myeloma or pancreatic cancer. In order to clarify this point, information regarding this higher rate of events in patients with pancreatic cancer has been added to the introduction section with the proposed references (page 1 Lines 38-40, references 6 and 7).

  1. In the introduction section, line 48, please add that the use of red cell growth factors has also been associated with an increased risk of VTE

We thank the reviewer for this remark. Red cell growth factor has been added with a related reference (Page 2, Lines 50-51) .

  1. In the introduction section, line 81, please add “pre-chemotherapy  haemoglobin or use of red cell growth factors’ in this item of the Khorana score

Thanks. We’ve added the use of red cell growth factors in the text (Page 3, Line 91).

  1. In the paragraph “use of DOACs for VTE prevention”, line 275 the authors cite the most recent guidelines on the management of VTE in cancer patients (ASCO and ITAC). Please add that the ITAC guidelines also recommend the use of primary thromboprophylaxis in patients with locally advanced or metastatic pancreatic cancer in the absence of a high risk of bleeding.

Thank you for this comment; a sentence regarding the specific recommendation for pancreatic cancer has been added to the section dedicated to the ITAC recommendations (Page 9, Lines 324-327). Moreover, an additional table summarizing the main recommendations for thromboprophylaxis has been added (table 6, page 9324-327)

  1. Horsted, F., J. West, and M.J. Grainge, Risk of venous thromboembolism in patients with cancer: a systematic review and meta-analysis. PLoS Med, 2012. 9(7): p. e1001275.

Round 2

Reviewer 1 Report

Acceptable in current form